# Protocol of efficacy of bifidobacteria intake on gastrointestinal symptoms in symptomatic type 2 diabetes mellitus patients in abdominis: An open-label, randomized controlled trial (Binary STAR study)

**Yoshitaka Hashimoto**[1,2]*, **Genki Kobayashi**[1], **Noriyuki Kitagawa**[1,3], **Hiroshi Okada**[1], **Masahide Hamaguchi**[1], **Michiaki Fukui**[1]

1 Department of Endocrinology and Metabolism, Kyoto Prefectural University of Medicine, Kyoto, Japan,
2 Department of Diabetes and Endocrinology, Matsushita Memorial Hospital, Moriguchi, Japan,
3 Department of Diabetology, Kameoka Municipal Hospital, Kameoka, Japan

* y-hashi@koto.kpu-m.ac.jp

**Data Availability Statement:** The datasets generated during and/or analyzed during the

## Abstract

### Background

This randomized, parallel-group study aims to investigate the effects of the probiotic *Bifidobacterium bifidum* G9-1 (BBG9-1) on the symptoms of diarrhea or constipation in patients with type 2 diabetes mellitus (T2DM).

### Methods

This study will examine 100 patients with T2DM who suffering from symptoms of diarrhea or constipation. Eligible patients will be randomly assigned 1:1 to two groups (group A, BBG9-1 group; group B, control group), after the baseline examination. Patients assigned to group A will receive probiotic BBG9-1 oral administration along with their current treatment for 12 weeks, and patients assigned to group B will continue the current treatment for 12 weeks without probiotic BBG9-1 oral administration. Subsequently, examinations similar to the baseline examinations will be performed. The primary endpoint will be a change in the Gastrointestinal Symptom Rating Scale (GSRS) total score from baseline to week 12. Secondary endpoints will include the following: change and percent change in parameters such as GSRS subdomain scores, fecal properties/Bristol stool form scale, defecation frequency, biomarkers, gut microbiota, and macronutrients and factors that affect GSRS total score or constipation/diarrhea subdomain scores from baseline to week 12.

### Discussion

The results of this study will clarify the utility of probiotic BBG9-1 in the treatment of diarrhea or constipation in patients with T2DM.

**Trial registration:** jRCTs051220127.

current study will not publicly available due to the lack of a statement in the study protocol enabling data sharing with a third party after the end of the study and in the informed consent documents as well as lack of approval for data sharing by an ethical review board and consent from the participants in this study. Thus, the ethical review board of Kyoto Prefecutral University not permitted. The datasets generated and/or analyzed during the current study will be available upon reasonable request. Contact information: Principal investigator, Michiaki Fukui, michiaki@koto.kpu-m.ac.jp Investigator, Yoshitaka Hashimoto, y-hashi@koto.kpu-m.ac.jp Department of Endocrinolgy and Metabolism, Kyoto Prefecutral University of Medicine, dmnaika@koto.kpu-m.ac.jp.

**Funding:** This study was supported by Biofermin Pharmaceutical Co., Ltd. Funding information is also upload at the "Source of Monetary Support / Secondary Sponsor" section in jRCT as the funding is provided by Biofermin Pharmaceuticals (https://jrct.niph.go.jp/en-latest-detail/jRCTs051220127).

**Competing interests:** Biofermin Pharmaceutical Co., Ltd. contributes to the research fund. The authors declare no conflicts of interest associated with this manuscript.

**Abbreviations:** BBG9-1, *Bifidobacterium bifidum* G9-1; T2DM, type 2 diabetes mellitus; GSRS, Gastrointestinal Symptom Rating Scale; ALT, alanine aminotransferase; AST, aspartate transaminase; CRF, case report form; FAS, full analysis set; PPS, per protocol set; SAE, serious adverse event.

# Introduction

## Background and rationale

The number of patients with type 2 diabetes mellitus (T2DM) is increasing worldwide [1]. In addition to microvascular and macrovascular complications, well-known complications of diabetes [1] and gastrointestinal diseases, such as reflux esophagitis, diarrhea, and constipation, have been reported to be associated with diabetes [2–4]. Diabetes medications reportedly increase the risk of gastrointestinal complications, such as constipation and diarrhea [5,6].

Recent studies have clarified the association between the gut microbiota and T2DM [7–11]. Dysbiosis causes a decrease in short-chain fatty acids and branched-chain amino acids, leading to chronic inflammation and insulin resistance [9]. On the other hand, it has been reported that probiotic bifidobacteria improves dysbiosis. Symptoms of diarrhea and constipation in individuals without diabetes have been improved by probiotic bifidobacteria [12–14]. In addition, we recently revealed that the probiotic *Bifidobacterium bifidum* G9-1 (BBG9-1) improved the symptoms of constipation and diarrhea in patients with T2DM [15]. However, the study was an open-label, single-arm exploratory study [15], and further research is needed to determine its effectiveness and mechanism. Therefore, this randomized, parallel-group study will investigate the efficacy of the probiotic BBG9-1 on the symptoms of diarrhea or constipation in patients with T2DM.

## Objectives

The purpose of this study was to evaluate the efficacy of probiotic BBG9-1 on the symptoms of diarrhoea or constipation in patients with T2DM using randomized, parallel-group study with a control group.

## Trial design

Open-label, randomized, multicenter, parallel-group study

# Materials and methods: Participants, interventions, and outcomes

## Study setting

This open-label, randomized controlled trial (Efficacy of **B**ifidobacteria **in**take on g**a**st**r**ointestinal **sy**mptoms in **S**ymptomatic **T**ype 2 diabetes mellitus patients in **A**bdominis: an open-label, **R**andomized controlled trial [**Binary STAR** study]) will be performed for 12 weeks in Japanese outpatients with T2DM at the Hospital of the Kyoto Prefectural University of Medicine (Kyoto, Japan), Matsushita Memorial Hospital (Moriguchi, Japan), and Kameoka Municipal Hospital (Kameoka, Japan). A total of 100 patients with T2DM, who have symptoms of diarrhea or constipation, will be included. This study was registered with the Japan Registry of Clinical Trials (jRCTs051220127) and approved by the ethics committees of the Kyoto Prefectural University of Medicine (CRB 2021028). This study will be conducted in accordance with the principles of the Declaration of Helsinki.

## Eligibility criteria

Patients must satisfy all of the following inclusion criteria. The inclusion criteria are as follows: (1) symptoms of diarrhea or constipation; (2) three or higher in Gastrointestinal Symptom Rating Scale (GSRS) subdomain score (diarrhea or constipation); (3) T2DM patients without diabetic polyneuropathy; (4) no new antibiotic usage within 12 weeks prior to consent; (5) no new diet interventions within 12 weeks prior to consent; (6) no changes in medications which

can affect gastrointestinal symptoms and microbiome (addition or discontinuation of medications or change in use or dosage of medications) within 12 weeks prior to consent; (7) aged 20 and up and under 75 years or less at the time of consent; (8) written consent obtained from the individual. The exclusion criteria are as follows: patients with (1) mean defecation frequency < 1 or > 42 times/ week within 4 weeks before giving their consent; (2) structural diseases, including inflammatory diseases (infectious enteritis, diverticulitis, Crohn's disease, and ulcerative colitis), and stenotic lesions, diagnosed by colonoscopy within 5 years before giving their consent; (3) having both ≥ 3 on the GSRS constipation subscale and > 4 on the Bristol stool scale; (4) having both ≥ 3 on the GSRS diarrhea subscale and < 4 on the Bristol stool scale; (5) gastrointestinal symptoms of both diarrhea and constipation; (6) past history of inflammatory bowel disease and celiac disease; (7) HbA1c level ≥ 9% at the time of consent; (8) use of alpha-glucosidase inhibitors within 4 weeks prior to consent; (9) cerebral infarction, stroke, or myocardial infarction within 12 weeks prior to consent; (10) severe hepatic dysfunction defined as alanine aminotransferase (ALT) or aspartate transaminase (AST) level 5 times equal or higher than the upper normal limit; (11) severe renal dysfunction defined as estimated glomerular filtration rate < 30 mL/min/1.73 m$^2$; (12) active malignant tumor; (13) allergy history of bifidobacteria; (14) usage of any other supplements or medications that affect intestinal function; (15) usage of medications that are likely to cause abdominal symptoms (e.g., gastro-prokinetic agents, drugs for dyspepsia, antiemetic agents, laxatives, and anticholinergics); (16) patients with changes in diet within 12 weeks prior to consent; (17) other conditions that the researcher or investigator thinks inappropriate for the study.

### Who will obtain informed consent?

Investigators or researchers will obtain written consent from the patients willing to participate in the trial. Patients will receive information sheets and discuss with investigators or researchers for the trial based on the information provided in the information sheets.

### Additional consent provisions for collection and use of participant data and biological specimens

If new information concerning this research is obtained, investigators and researchers will provide additional explanations to the participants and revise the consent document, as necessary.

### Interventions

**Explanation for the choice of comparators.** In this trial, no medication will be administered to the control group.

**Intervention description.** A flowchart for enrolment and follow-up visits is shown in Fig 1. Written informed consent and provisional registration will be obtained from all participants. Within 8 weeks of provisional registration, baseline examinations will be performed, and participants will be registered based on the aforementioned inclusion and exclusion criteria. Eligible patients will then be randomized into two groups (group A, BBG9-1 group; group B, control group) using the minimization method. Among the background factors of the study participants, age (<65 years or ≥65 years), sex, and the abdominal symptoms (constipation or diarrhea) are used as allocation factors. Patients assigned to group A will receive probiotic BBG9-1 oral administration (Biofermin® tablets containing 12 mg of bifidobacteria, 2 tablets 3 times a day) along with their current treatment for 12 (± 3) weeks, and patients assigned to group B will continue the current treatment for 12 (± 3) weeks without probiotic BBG9-1 oral administration. Subsequently, the examinations, which are the same as the baseline examinations, will be performed 12 (± 3) weeks after the baseline examinations. For patients assigned

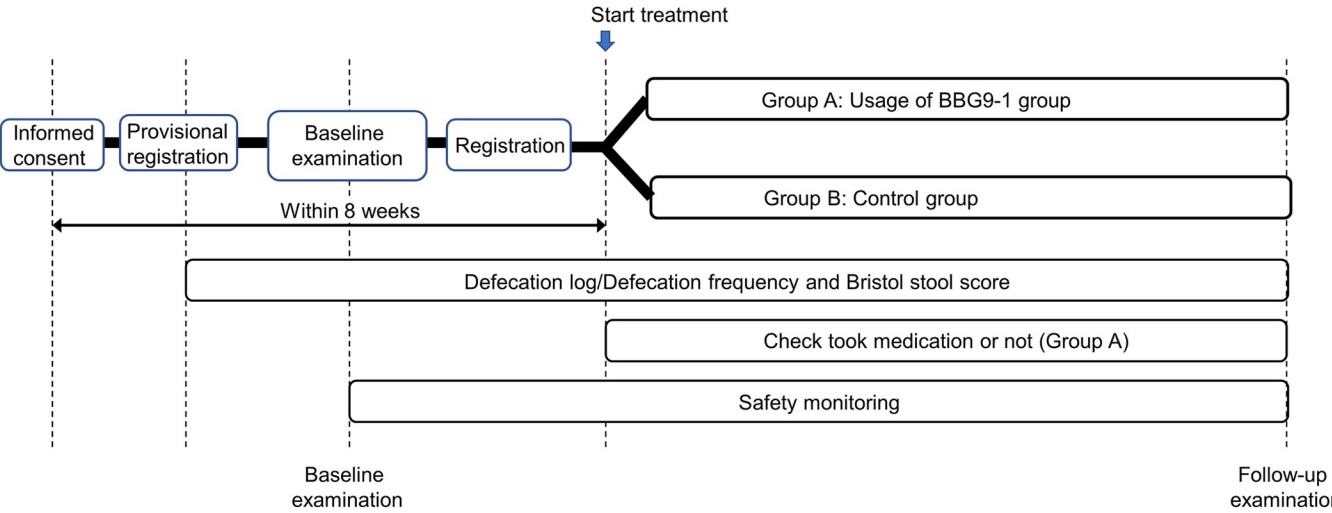

**Fig 1. Study design of Binary STAR study.**

to group A, the date of initiation of medication will be the starting date of observation for the study. For patients assigned to group B, the date of sampling at the baseline examination will be the starting date of observation for the study. Research patients who are not enrolled in the study will be excluded from participation at the time of their failure to enroll. Such cases will not be counted as the target number of cases to be enrolled in the study.

**Criteria for discontinuing or modifying allocated interventions.** The criteria for discontinuation of observation are as follows: (1) failure to register; (2) worse glycemic control (HbA1c $\geq$ 10%); (3) withdrawal of consent; (4) serious incompatibility after registration; (5) discontinuation of medications because of worsening of disease and/or complications; (6) discontinuation of the trial because of illness or adverse events; (7) pregnancy; (8) significant non-adherence ($<$ 75% of the number of tablets each participant plans to take internally or $>$ 120%); (9) significant deviation from the research protocol; (10) investigator/researcher's determination to stop the study for other reasons. The data shall be fixed by the Data Handling Committee after handling in a blinded manner. Even if there is a deviation from the research protocol, observations shall be continued as much as possible.

The criteria for discontinuation of the trial are as follows: (1) significant information regarding the quality, safety, or efficacy of Biofermin® tablets is obtained; (2) recruitment of research participants is difficult, and there is a large discrepancy from the planned number of cases; (3) the objective of the research is achieved before the planned number of cases or planned duration is reached; (4) instructions to change the research plan, and it is difficult to accept these instructions.

**Strategies to improve adherence to interventions.** Adherence will be assessed by having the patients maintain a daily log of medication during the observation period.

**Relevant concomitant care permitted or prohibited during the trial.** During the study period, changes in medications other than the addition of BBG9-1 to the intervention group will not be permitted except in unavoidable circumstances.

**Provisions for post-trial care.** Post-trial care is not planned. The possible risks are detailed in the informed consent form. If any patient harmed during the trial, appropriate medical and nursing care will be provided.

## Outcomes

The primary endpoint of this trial will be the change in the total GSRS score from baseline to follow-up examination (12 ± 3 weeks after baseline examination). Secondary endpoints of this trial will be as follows: (1) change and percent change in following parameters from baseline to follow-up examination: GSRS subdomain scores, fecal properties/Bristol stool form scale, defecation frequency, blood test biomarkers (total bile acid, glycoalbumin, glucagon-like peptide-1, glucose-dependent insulinotropic polypeptide, glucagon, HbA1c, fasting plasma glucose, C-peptide, AST, ALT, gamma-glutamyltransferase, total cholesterol, high-density lipoprotein cholesterol, low-density lipoprotein cholesterol, triglycerides, blood urea nitrogen, and creatinine), physical examination (body weight and body mass index), urinary albumin creatinine ratio, gut microbiota (type of enterobacteria, relative abundance, alpha diversity, and beta diversity), macronutrients (total energy, carbohydrate, fat, protein, and dietary fiber intakes measured by brief-type self-administered diet history questionnaire [BDHQ]) [16]; (2) proportion of participants whose fecal property of Bristol stool form scale is within normal range (3.5–4.5); (3) medication adherence to the BBG9-1; (4) factors that affect GSRS total score or constipation/diarrhea subdomain scores; (5) stratification of following items by baseline symptoms (constipation/diarrhea): GSRS total score, GSRS subdomain scores, proportion of participants whose fecal properties are within normal range, defecation frequency, blood test biomarkers, and gut microbiota. The exploratory endpoints of this trial will be as follows: (1) blood, urine, and stool metabolomes (short, medium, and long chain fatty acids, amino acids, carbohydrates, and bile acids); (2) items other than nutrients evaluated in the secondary endpoints and items evaluated by the diabetes treatment satisfaction questionnaire (DTSQ) [17]; (3) items other than physical examination evaluated in the secondary endpoints; (4) items other than gut microbiota evaluated in the secondary endpoints; (5) change in lipopolysaccharide-binding protein; (6) subgroup analysis by age, sex, etc. Harm outcomes will also be evaluated based on various laboratory abnormalities and adverse reactions to the medication.

## Participant timeline

The schedule of enrolment, interventions, and visits for participants is shown in Figs 1 and 2.

## Sample size

In a sub-analysis of a previous study [15,18], BBG9-1 oral administration changed the total GSRS score by -0.441 ± 0.499. In the non-treatment period, the change in the total GSRS score was -0.197 ± 0.314. Based on these findings, assuming a difference of 0.244 ± 0.412 in the change in GSRS between the two groups in the study, we obtained an alpha error of 0.05, a power of 0.80, and a sample size ratio of 1:1; 45 cases per group would be needed to detect a significant between-group difference. Dropouts with a dropout rate of 10%, 50 cases in one group, and 100 cases in both groups have been set as the target number of cases for this study.

## Recruitment

Outpatients with T2DM at the Department of Endocrinology and Diabetes of Kyoto Prefectural University of Medicine Hospital (Kyoto, Japan), Department of Diabetes of Kameoka Municipal Hospital (Kameoka, Japan), and Department of Diabetes and Endocrinology of Matsushita Memorial Hospital (Moriguchi, Japan) will be recruited to participate in this study.

| | STUDY PERIOD | | | | |
|---|---|---|---|---|---|
| | Enrolment | Allocation | Post-allocation | | Close-out |
| TIMEPOINT | -8 to 0 weeks | 0 | -8 to 0 weeks | 12 ± 3 weeks | |
| **ENROLMENT:** | | | | | |
| Eligibility screen | X | | | | |
| Informed consent | X | | | | |
| Allocation | | X | | | |
| **INTERVENTIONS:** | | | | | |
| BBG9-1 group | | | ●━━━━━━━● | | |
| Control group | | | | | |
| **ASSESSMENTS:** | | | | | |
| HbA1c, Age, sex, GSRS, AST,ALT, estimated glomerular filtration rate, mediation usage history | X | X | | | |
| Bristol stool scale | ●━━━━━● | | | | |
| GSRS, blood test biomarkers, physical examination, urinary albumin creatinine ratio, gut microbiota, macronutrients, metabolomes of blood, urine, and stool, diabetes treatment satisfaction questionnaire | | | X* | X | X |
| Bristol stool scale | | | ●━━━━━━━● | | |
| Adherence to medication | | | ●━━━━━━━● | | |

**Fig 2. Schedule of enrolment, interventions, and assessments.** *, Must be performed prior to initiation of the *Bifidobacterium bifidum* G9-1. ALT, alanine aminotransferase; AST, aspartate transaminase; BBG9-1, *Bifidobacterium bifidum* G9-1; GSRS, Gastrointestinal Symptom Rating Scale.

## Assignment of interventions: Allocation

**Sequence generation.** The investigators or researchers will enter the necessary information into the web system; if eligible, based on the GSRS score and the Bristol scale score, perform the main enrolment/assignment, and confirm the results. The participants will be assigned to group A or group B using the minimization method, with age, sex, and type of symptom (constipation or diarrhea) as allocation factors.

**Implementation.** The researchers will enter the necessary information into the web system and will check the allocation.

## Data collection and management

**Plans for assessment and collection of outcomes.** The researchers will manage the data of the participants using the central registry number assigned by the data center. In this study, data on study participants reported by the researchers will be collected at each observation point using a case report form (CRF), survey forms (GSRS, BDHQ, and DTSQ), logbooks

(defecation frequency, stool characteristics (Bristol scale score), and adherence to BBG9-1). We will use the average of the 1-week Bristol scale scores as the Bristol scale score. The survey forms of 12-week after the baseline examination and logbooks during the observation periods will be sealed using a special envelope and sent to the data center in such a way that the investigators or researchers cannot confirm the contents of the responses. The investigators or researchers will ensure that the survey forms and logbooks other than those answered and recorded by the research participants themselves are complete in order to ensure that the data recorded by the research participants (CRFs) are not lost or destroyed. If any data correction is necessary, the date and reason for the correction shall be recorded. Researchers shall confirm that all data, including the corrected data, are accurate and complete and shall ensure the quality of the data. For data submitted in the CRF, the data management staff will maintain a central registry of all data from the site.

**Data management.** The data management for this study will be conducted in accordance with the procedures and checklists. The study participants will be identified using their central registry numbers. The electronic transfer of data to the research participants must be approved by the data manager. When data are transferred over an unsecured electronic network, they must be encrypted at source. For the discontinued cases, data will be collected up to the point of discontinuation.

**Confidentiality.** Information that can identify research participants (e.g., name, address, and telephone number) will not be gathered or recorded in the CRF. Data management staff will use the central registration number when making inquiries to medical institutions.

**Plans for collection, laboratory evaluation, and storage of biological specimens for genetic or molecular analysis for this trial/future use.** Laboratory blood tests will be performed according to local clinical procedures, using standard and sterilized vacuum tubes for serum separation for C peptide, AST, ALT, gamma-glutamyltransferase, total cholesterol, high-density lipoprotein cholesterol, low-density lipoprotein cholesterol, triglyceride, blood urea nitrogen, creatinine, total bile acid, glycoalbumin; EDTA-2Na with NaF and fetal bovine serum for HbA1c; BD® P800 for glucagon-like peptide-1, glucose-dependent insulinotropic polypeptide, and glucagon. Furthermore, lipopolysaccharide-binding protein will be measured using enzyme-linked immunosorbent assay. Urinary laboratory tests will also be performed according to local clinical procedures. Gut microbiota will be measured using shotgun metagenomic sequencing. Metabolomes of serum, urine, and stool will also be measured using gas chromatography-tandem mass spectrometry.

## Statistical methods

**Statistical methods for primary and secondary outcomes.** Full analysis set (FAS) analysis will be performed for the primary and secondary endpoints. If necessary, per protocol set (PPS) analysis will be performed as a sensitivity analysis for the FAS analysis. In principle, the analysis should be performed at a two-sided significance level of 5%. The person responsible for the statistical analysis will prepare a separate statistical analysis plan that specifies the details of the statistical methods, including data handling. A statistical analysis plan will be prepared before the data are fixed. The primary endpoints will be treated as validation results, whereas the other endpoints will be treated as exploratory results.

The FAS will include the participants who are enrolled in the study and assigned to the study treatment. However, data on participants who violated the research protocol, such as enrolment without consent and registration outside the contract period, will be excluded. The PPS will include research participants who will be excluded from the FAS for the following reasons: violation of eligibility criteria, violation of exclusion criteria, violation of concomitant

use of prohibited medications, and medication compliance rate > 120% or < 75%. The safety analysis set will include participants in whom research treatment has been initiated.

Summary statistics for the background data of study participants will be calculated for each group. Comparisons between groups will be made using the χ-square test or Fisher's direct probability test for categorical variables and the two-sample t-test or Wilcoxon rank-sum test for continuous variables.

The primary endpoints will be evaluated using analysis of covariance. A two-sample t-test will be performed for between-group comparisons and a one-sample t-test will be performed for within-group comparisons. The secondary endpoints will be evaluated using a two-sample t-test or one-sample t-test. For data without a normal distribution, the log-transformation will be attempted; however, if it does not improve the distribution, Wilcoxon rank-sum test or Wilcoxon signed-rank will be performed. For the Bristol scale score, between-group comparisons will be performed using the Wilcoxon rank-sum test, and within-group comparisons will be performed using the Wilcoxon signed-rank test. For percentage endpoints, the χ-square test or Fisher's direct probability test will be performed. For safety, a list of all adverse events will be prepared for each group in the safety analysis population; if necessary, comparisons between groups will be made using Fisher's direct probability test.

## Interim analyses

No interim analyses will be conducted in this study.

## Methods for additional analyses (e.g., subgroup analyses)

Subgroup analyses of baseline symptoms (constipation/diarrhea), GSRS total score, GSRS subdomain scores, proportion of participants whose fecal properties are within the normal range, defecation frequency, blood test biomarkers, and gut microbiota will be performed.

## Methods of analysis to handle protocol nonadherence and any statistical methods to handle missing data

Cases of violation of the concomitant use of prohibited medications will be included in the FAS but not in the PPS. Postprandial data of blood glucose, C-peptide, active glucagon-like peptide-1, and total bile acids are not included in the FAS and PPS because they are significantly less reliable. Missing data are treated as missing data and will not be complemented.

## Plans to provide access to the full protocol, participant-level data, and statistical code

Access to the datasets used and analyzed during the study is available in a fully anonymized form from the sponsor upon reasonable request.

## Oversight and monitoring

**Composition of the coordinating center and trial steering committee.** The research office will assist the principal investigator in the management and supervision of the study from medical, scientific, and ethical perspectives and will liaise with the parties involved in the conduct of the study.

**Composition of the data monitoring committee, its role, and reporting structure.** The data monitoring committee consists of research investigators and statistical analysts. The handling of data deviating from the study design and the handling of missing data will be evaluated in a blinded manner prior to analysis to determine the handling of all data.

## Reporting of adverse events and harms

The incidence of diseases and adverse events will be used as a safety measure. These events will be recorded using CRF throughout the observation period. The analysis will be conducted using a safety analysis population after data fixation. Any unfavorable medical event that occurs to a research participant during the course of this study, including an exacerbation of previous illnesses, will be treated as an "adverse event." Worsening of diabetes-related macrovascular and microvascular complications at the time of initiation of medication will also be treated as an adverse event. In this study, if the data related outcomes, including primary endpoints, secondary endpoints and exploratory endpoints, worsens, it will not be treated as an adverse event. When an adverse event will be observed, researchers will immediately take appropriate measures and report necessary information. If administration of BBG9-1 is discontinued or treatment for an adverse event becomes necessary, the researchers will inform the research participant.

If an adverse event occurs, the name, timing, degree, severity, outcome, date of confirmation of the outcome, relevance to the study, reason for judgment, suspect medication(s), treatment of the suspect medication(s), and other treatment details shall be recorded in the CRF. In principle, symptoms observed during the study will be followed-up until they normalize or recover to a level not considered an adverse event, or until symptoms stabilize or will be fixed in the case of irreversible adverse events due to organic damage.

The degree of adverse events will be classified according to the following criteria: mild, not detrimental to daily life; moderate, interference with daily life but only to the extent that activities can be performed with considerable patience; severe, to the extent that it significantly interferes with the performance of daily activities.

The outcome of adverse events will be classified according to the following criteria: recovery, recovery to a level that is normalized or not considered an adverse event; slightly relieved, symptoms have lessened but have not normalized; unrecovered, symptoms have not yet normalized at the time of the event; recovered but with residual aftereffects, recovered but still dysfunctional to the extent that it interferes with daily life; death; unknown, the outcome is unknown due to interruption of hospital visits.

Relevance to this study will be judged "not related" or "related" according to the following relevance criteria: the condition of the research participant; the time relationship with the treatment; the possibility of other factors.

A serious adverse event (SAE) is defined as any unwanted or unintended sign, symptom, or illness for a research participant, which is causally related to the research drug and corresponds to the following: 1) death, 2) life-threatening, 3) requires hospitalization or prolonged hospitalization for treatment, 4) results in permanent or significant disability or dysfunction, 5) the patient suffers from any other medically significant condition, 6) cases that are as serious as those listed in 1) to 5), and 7) congenital disease or abnormality in later generations.

If SAE occurs, the researchers will promptly take the necessary treatment, and regardless of the causal relationship between the SAE and the investigational drug in this study, the researchers shall report to the data center within 15 calendar days of becoming aware of the SAE, in accordance with the procedures of the institution. The data center will immediately summarize and report it to the administrator of the medical institution. The data center will immediately compile information, obtain instructions from the researchers, and report it to the research fund contributor (Biofermin Pharmaceutical Co., Ltd.). In the event of an adverse event other than an SAE, an adverse event will be noted in the CRF and reported to the data center upon submission of the CRF. The data center shall compile such information as appropriate, report it to the investigators and researchers, and receive corresponding instructions.

The data center will also report the information to the research fund contributor (Biofermin Pharmaceutical Co., Ltd.), as appropriate. If new information concerning this research is obtained, the researchers will provide additional explanations to the participants and revise the consent and explanation documents as necessary.

### Frequency and plans for auditing trial conduct

The auditing trial conduct will be conducted by EviPRO, Inc., in accordance with the procedures.

### Plans for communicating important protocol amendments to relevant parties (e.g., trial participants and ethical committees)

Important protocol modifications that may have an impact on the potential benefit for the participant, the safety of participant, or the conduct of the study, will be communicated to the trial registries, ethical committee, sponsor, and trial participants.

### Dissemination plans

Before enrolment of the first study participants, the study plan will be registered and published in the database maintained by the Ministry of Health, Labour, and Welfare (Japan Registry of Clinical Trials/jRCT). The progress of the research will be updated as appropriate, and completion of the research will be reported without delay. All data and results obtained in this study will belong to the investigators and researchers. These results will be published in scientific journals.

## Discussion

This randomized, parallel-group, comparative study explores the effects of the probiotic BBG9-1 on the symptoms of diarrhea or constipation in patients with T2DM.

Previous studies have revealed that many patients with diabetes have gastrointestinal diseases such as diarrhea and constipation [2–4]. In addition, medications for diabetes increase the risk of gastrointestinal complications [5]. Furthermore, recent studies have revealed a relationship between gastrointestinal complications and gut microbiota [6]. Therefore, treatment to improve abdominal symptoms, such as diarrhea and constipation, through the improvement of the gut microbiota is needed.

Recent studies have revealed that probiotics containing BBG9-1 improve the balance of gut microbiota and result in improved gastrointestinal complications in both animal models [19] and humans [15]. However, because previous human studies were conducted in a single group without a control group, further investigation is needed to clarify the efficacy and mechanism of BBG9-1 in patients with T2DM having constipation or diarrhea.

This is a randomized, parallel-group, comparative study with a control group to evaluate the effect of the probiotic BBG9-1 on improving abdominal symptoms in patients with T2DM. The results of this study will clarify the efficacy of the probiotic BBG9-1 in the treatment of diarrhea or constipation in patients with T2DM.

## Supporting information

**S1 Checklist. SPIRIT 2013 checklist: Recommended items to address in a clinical trial protocol and related documents\*.**
(DOC)

## Author Contributions

**Conceptualization:** Yoshitaka Hashimoto, Michiaki Fukui.

**Data curation:** Yoshitaka Hashimoto, Genki Kobayashi, Noriyuki Kitagawa, Hiroshi Okada, Masahide Hamaguchi, Michiaki Fukui.

**Funding acquisition:** Yoshitaka Hashimoto, Michiaki Fukui.

**Investigation:** Yoshitaka Hashimoto.

**Project administration:** Yoshitaka Hashimoto, Michiaki Fukui.

**Writing – original draft:** Yoshitaka Hashimoto, Genki Kobayashi.

**Writing – review & editing:** Noriyuki Kitagawa, Hiroshi Okada, Masahide Hamaguchi, Michiaki Fukui.

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
