## [Decision Letter · Decision Letter 0]

10 Oct 2023

PONE-D-23-03350Protocol of efficacy of bifidobacteria intake on gastrointestinal symptoms in symptomatic type 2 diabetes mellitus patients in abdominis: an open-label, randomized controlled trial (Binary STAR study)PLOS ONE

Dear Dr. Hashimoto,

Thank you for submitting your manuscript to PLOS ONE. After careful consideration, we feel that it has merit but does not fully meet PLOS ONE’s publication criteria as it currently stands. Therefore, we invite you to submit a revised version of the manuscript that addresses the points raised during the review process.

ACADEMIC EDITOR: The manuscript needs minor revision.

We look forward to receiving your revised manuscript.

Kind regards,

Gang Qin, PhD, MD

Academic Editor

PLOS ONE

Manuscript Number:

- 10.3164/jcbn.20-100

In your revision ensure you cite all your sources (including your own works), and quote or rephrase any duplicated text outside the methods section. Further consideration is dependent on these concerns being addressed.

3. In the competing interests statement within the manuscript and in the online submission form, please declare your affiliation with Biofermin Pharmaceutical Co. and thoroughly report any potential competing interests related to this affiliation.

a) If there are ethical or legal restrictions on sharing a de-identified data set, please explain them in detail (e.g., data contain potentially sensitive information, data are owned by a third-party organization, etc.) and who has imposed them (e.g., an ethics committee). Please also provide contact information for a data access committee, ethics committee, or other institutional body to which data requests may be sent. Please note that authors, including Corresponding Authors, are not permitted to be the sole point of contact for data requests.

b) If there are no restrictions, please provide the minimal anonymized data set necessary to replicate your study findings as either Supporting Information files or to a stable, public repository and provide us with the relevant URLs, DOIs, or accession numbers. For a list of acceptable repositories, please see http://journals.plos.org/plosone/s/data-availability#loc-recommended-repositories.

7. Please include a caption for figure 2.

8. We note that the original protocol file you uploaded contains a confidentiality notice indicating that the protocol may not be shared publicly or be published. Please note, however, that the PLOS Editorial Policy requires that the original protocol be published alongside your manuscript in the event of acceptance. Please note that should your paper be accepted, all content including the protocol will be published under the Creative Commons Attribution (CC BY) 4.0 license, which means that it will be freely available online, and any third party is permitted to access, download, copy, distribute, and use these materials in any way, even commercially, with proper attribution.

Therefore, we ask that you please seek permission from the study sponsor or body imposing the restriction on sharing this document to publish this protocol under CC BY 4.0 if your work is accepted. We kindly ask that you upload a formal statement signed by an institutional representative clarifying whether you will be able to comply with this policy. Additionally, please upload a clean copy of the protocol with the confidentiality notice (and any copyrighted institutional logos or signatures) removed.

Additional Editor Comments:

The manuscript needs minor revision.

Reviewers' comments:

Reviewer's Responses to Questions

**Comments to the Author**

1. Does the manuscript provide a valid rationale for the proposed study, with clearly identified and justified research questions?

Reviewer #1: Yes

Reviewer #2: Yes

2. Is the protocol technically sound and planned in a manner that will lead to a meaningful outcome and allow testing the stated hypotheses?

Reviewer #1: Yes

Reviewer #2: Yes

3. Is the methodology feasible and described in sufficient detail to allow the work to be replicable?

Reviewer #1: Yes

Reviewer #2: Yes

4. Have the authors described where all data underlying the findings will be made available when the study is complete?

Reviewer #1: Yes

Reviewer #2: Yes

5. Is the manuscript presented in an intelligible fashion and written in standard English?

Reviewer #1: Yes

Reviewer #2: Yes

6. Review Comments to the Author

You may also provide optional suggestions and comments to authors that they might find helpful in planning their study.

Reviewer #1: page 8: just clarifying: 6 tables 3 times a day is 18 tablets in total for the 12 (+/- 3) weeks? What happens if the patient misses 1 tablet or 6 tablets or 18 tablets?

page 9: line 164 - How is the 75% calculated for patients that only have 9 weeks of treatment (i.e. if they take it for 8.5 weeks that is less than 75% of 12 weeks, but more than 75% of 9 weeks)

line 183: what does "in principle" mean here? Are changes allowed or not?

line 192: baseline to week 12 - what if patient's only have 9 weeks of treatment?

page 16: line 319 "The FAS will include the participants who are enrolled in the study and assigned to the study treatment. However, data on participants who violated the research protocol will be excluded." Doesn't this make the FAS the same as the per protocol set? There is a definition provided for PPS - but please clarify who is in the FAS.

page 19 line 384: "Worsening of a complication at the time of initiation of medication will also be treated as an adverse event" - isn't this covered by the previous sentence of "an exacerbation of a preexisting condition"?

line 385: "If the measurement in this study worsens, it will not be treated as an adverse event." does this refer to constipation and diarrhoea only? Please clarify.

Table 1 and Figure 2 contain very similar information. Are both required?

Reviewer #2: Thanks for inviting me to review the paper. "Protocol of efficacy of bifidobacteria intake on gastrointestinal symptoms in

symptomatic type 2 diabetes mellitus patients in abdominis: an open-label, randomized controlled trial (Binary STAR study)" overall, the protocol is clear and well written. The study design is clearly presented with most of the related details. A few comments are here to improve the manuscript.

1, For the Objectives in the main text, would it be better to say the aim of this paper (show the details of study design), rather than the aim of the trial?

2, More information may be needed for the method of randomization, expend a bit for the minimization?

3, For the intervention, is it required that probiotic BBG9-1 should be taken with water under 37 °c?

7. PLOS authors have the option to publish the peer review history of their article (what does this mean?). If published, this will include your full peer review and any attached files.

Reviewer #1: No

Reviewer #2: **Yes: **Xin Liu

---

## [Author Response · Author response to Decision Letter 0]

4 Nov 2023

Prof. Gang Qin, 

Academic Editor

PLOS ONE

Dear Editors:

Thank you for your kind letter concerning our manuscript on Oct 10, 2023.

Enclosed please find our revised manuscript entitled " Protocol of efficacy of bifidobacteria intake on gastrointestinal symptoms in symptomatic type 2 diabetes mellitus patients in abdominis: an open-label, randomized controlled trial (Binary STAR study)”, manuscript ID of which is PONE-D-23-03350.

At first, we would like to thank associate editor and reviewers for constructive comments on our manuscript.

According to the associate editor and reviewers’ comments, we have carefully revised our manuscript. Responses to associate editor and reviewers’ comments are described as below.

Yours faithfully,

Yoshitaka Hashimoto, MD, PhD

Department of Diabetes and Endocrinology, Matsushita Memorial Hospital

Address: 5-55 Sotojima-cho, Moriguchi 570-8540, Japan

Fax: +81669924845

Tel: +81669921231

E-mail: y-hashi@koto.kpu-m.ac.jp

 

Response to Associate Editors comments:

Response

Thank you for your comment. According to your comment, we have revised our manuscript.

- 10.3164/jcbn.20-100

In your revision ensure you cite all your sources (including your own works), and quote or rephrase any duplicated text outside the methods section. Further consideration is dependent on these concerns being addressed.

Response

Thank you for your suggestion. According to your suggestion, we have revised our manuscript. 

3. In the competing interests statement within the manuscript and in the online submission form, please declare your affiliation with Biofermin Pharmaceutical Co. and thoroughly report any potential competing interests related to this affiliation.

Response

Thank you for your comment. Biofermin Pharmaceutical Co., Ltd. contributes to the research fund. The authors declare no conflicts of interest associated with this manuscript. According to your comment, we have revised “Competing interest” section described as below. The authors declare no conflicts of interest associated with this manuscript.

“Biofermin Pharmaceutical Co., Ltd. contributes to the research fund. The authors declare no conflicts of interest associated with this manuscript.”

Response

Thank you for your comment. According to your comment, we have revised matched the ‘Funding Information’ and ‘Financial Disclosure’ sections. Since this study is not the award-winning study, there is no grant number. 

a) If there are ethical or legal restrictions on sharing a de-identified data set, please explain them in detail (e.g., data contain potentially sensitive information, data are owned by a third-party organization, etc.) and who has imposed them (e.g., an ethics committee). Please also provide contact information for a data access committee, ethics committee, or other institutional body to which data requests may be sent. Please note that authors, including Corresponding Authors, are not permitted to be the sole point of contact for data requests.

b) If there are no restrictions, please provide the minimal anonymized data set necessary to replicate your study findings as either Supporting Information files or to a stable, public repository and provide us with the relevant URLs, DOIs, or accession numbers. For a list of acceptable repositories, please see http://journals.plos.org/plosone/s/data-availability#loc-recommended-repositories.

Response

Thank you for your comment. According to your comment, we have revised the Availability of data and materials section and added this point in the cover letter. 

“Availability of data and materials: 

The minimum anonymized dataset necessary to reproduce the study results will be attached as a Supporting Information file.”

Response

Thank you for your comment. According to your comment, we have revised the manuscripts.

7. Please include a caption for figure 2.

Response

Thank you for your suggestion. According to reviewer 1’s comment, we have removed Figure 2. 

8. We note that the original protocol file you uploaded contains a confidentiality notice indicating that the protocol may not be shared publicly or be published. Please note, however, that the PLOS Editorial Policy requires that the original protocol be published alongside your manuscript in the event of acceptance. Please note that should your paper be accepted, all content including the protocol will be published under the Creative Commons Attribution (CC BY) 4.0 license, which means that it will be freely available online, and any third party is permitted to access, download, copy, distribute, and use these materials in any way, even commercially, with proper attribution.

Therefore, we ask that you please seek permission from the study sponsor or body imposing the restriction on sharing this document to publish this protocol under CC BY 4.0 if your work is accepted. We kindly ask that you upload a formal statement signed by an institutional representative clarifying whether you will be able to comply with this policy. Additionally, please upload a clean copy of the protocol with the confidentiality notice (and any copyrighted institutional logos or signatures) removed.

Response

Thank you for your comment. After accepting the manuscript, we agree that all content including the protocol will be published under the Creative Commons Attribution (CC BY) 4.0 license and we upload a clean copy of the protocol.

Response

Thank you for your comment. According to your comment, we have checked the references.

 

Response to Reviewer 1:

1. page 8: just clarifying: 6 tables 3 times a day is 18 tablets in total for the 12 (+/- 3) weeks? What happens if the patient misses 1 tablet or 6 tablets or 18 tablets?

Response

Thank you for your comment and sorry for bothering you. In this study, the participants will be take 2 tables 3 times a day and 6 tables in total for the 12 (+/- 3) weeks. According to your comments, we have revised Intervention description section, described as below. 

“Patients assigned to group A will receive probiotic BBG9-1 oral administration (Biofermin® tablets containing 12 mg of bifidobacteria, 2 tables 3 times a day) along with their current treatment for 12 (± 3) weeks,”

2. page 9: line 164 - How is the 75% calculated for patients that only have 9 weeks of treatment (i.e. if they take it for 8.5 weeks that is less than 75% of 12 weeks, but more than 75% of 9 weeks)

Response

Thank you for your comment. In this study, significant nonadherence is defined as less than 75% of the total number of doses in each participant. For participants receiving 9 weeks of treatment, the number of tablets they would have taken during the 9-week treatment period will be used as the basis for determining the number of tablets they would have taken. According to your comment, we have revised the Criteria for discontinuing or modifying allocated interventions section, described as below.

“(8) significant nonadherence (< 75% of the number of tablets each participant plans to take internally or > 120%);”

3. line 183: what does "in principle" mean here? Are changes allowed or not?

Response

Thank you for your comment. During the study period, changes will not allow except in unavoidable circumstances. According to your comment, we have revised the Relevant concomitant care permitted or prohibited during the trial section, described as below.

“During the study period, changes in medications other than the addition of BBG9-1 to the intervention group will not be permitted except in unavoidable circumstances.”

4. line 192: baseline to week 12 - what if patient's only have 9 weeks of treatment?

Response

Thank you for your comment. In this study, 12 ± 3 weeks of treatment will be allowed. According to your comment, we have revised the Outcomes section, described as below.

“The primary endpoint of this trial will be the change in the total GSRS score from baseline to follow-up examination (12 ± 3 weeks after baseline examination). Secondary endpoints of this trial will be as follows: (1) change and percent change in following parameters from baseline to follow-up examination.”

5. page 16: line 319 "The FAS will include the participants who are enrolled in the study and assigned to the study treatment. However, data on participants who violated the research protocol will be excluded." Doesn't this make the FAS the same as the per protocol set? There is a definition provided for PPS - but please clarify who is in the FAS.

Response

Thank you for your comment. In this study, the FAS will include the participants who are enrolled in the study and assigned to the study treatment. However, data on participants who violated the research protocol, such as enrollment without consent and registration outside the contract period, will be excluded. According to your comment, we have revised the Statistical methods for primary and secondary outcomes section, described as below.

“The FAS will include the participants who are enrolled in the study and assigned to the study treatment. However, data on participants who violated the research protocol, such as enrollment without consent and registration outside the contract period, will be excluded.”

6. page 19 line 384: "Worsening of a complication at the time of initiation of medication will also be treated as an adverse event" - isn't this covered by the previous sentence of "an exacerbation of a preexisting condition"?

Response

Thank you for your comment. In this study, complications mean diabetes-related macrovascular and microvascular complications and preexisting condition means previous illnesses. According to your comment, we have revised the Reporting of adverse events and harms section, described as below.

“Any unfavorable medical event that occurs to a research participant during the course of this study, including an exacerbation of previous illnesses, will be treated as an “adverse event.” Worsening of diabetes-related macrovascular and microvascular complications at the time of initiation of medication will also be treated as an adverse event.”

7. line 385: "If the measurement in this study worsens, it will not be treated as an adverse event." does this refer to constipation and diarrhea only? Please clarify.

Response

Thank you for your comment. In this study, if the data related outcomes, including primary endpoints, secondary endpoints and exploratory endpoints, worsens, it will not be treated as an adverse event. According to your comment, we have revised the Reporting of adverse events and harms section, described as below.

“In this study, if the data related outcomes, including primary endpoints, secondary endpoints and exploratory endpoints, worsens, it will not be treated as an adverse event.”

8. Table 1 and Figure 2 contain very similar information. Are both required?

Response

Thank you for your suggestion. As you say, the contents of Table 1 and Figure 2 are similar. According to your suggestion, we have removed Figure 2. 

Response to Reviewer 2

1. For the Objectives in the main text, would it be better to say the aim of this paper (show the details of study design), rather than the aim of the trial?

Response

Thank you for your suggestion. As you say, it would be better to show the aim of this study (show the details of study design), rather than the aim of the study. According to your suggestion, we have revised the Objectives section, described as below. 

“The purpose of this study was to evaluate the efficacy of probiotic BBG9-1 on the symptoms of constipation or diarrhea in patients with T2DM using randomized, parallel-group study with a control group.”

2. More information may be needed for the method of randomization, expend a bit for the minimization?

Response

Thank you for your suggestion. In this study, eligible patients will then be randomized into two groups (group A, BBG9-1 group; group B, control group) using the minimization method. Among the background factors of the study participants, age (<65 years or ≥65 years), sex, and the abdominal symptoms (constipation or diarrhea) are used as allocation factors. According to your suggestion, we have revised the Intervention description section, described as below.

“Eligible patients will then be randomized into two groups (group A, BBG9-1 group; group B, control group) using the minimization method. Among the background factors of the study participants, age (<65 years or ≥65 years), sex, and the abdominal symptoms (constipation or diarrhea) are used as allocation factors.”

3. For the intervention, is it required that probiotic BBG9-1 should be taken with water under 37 °c?

Response

Thank you for your comment. In this study, probiotic BBG9-1 does not necessarily need to be taken with water under 37°c. However, it is recommended to be taken with water under 37°c

---

## [Decision Letter · Decision Letter 1]

5 Dec 2023

Protocol of efficacy of bifidobacteria intake on gastrointestinal symptoms in symptomatic type 2 diabetes mellitus patients in abdominis: an open-label, randomized controlled trial (Binary STAR study)

PONE-D-23-03350R1

Dear Dr. Hashimoto,

We’re pleased to inform you that your manuscript has been judged scientifically suitable for publication and will be formally accepted for publication once it meets all outstanding technical requirements.

Kind regards,

Gang Qin, PhD, MD

Academic Editor

PLOS ONE

Additional Editor Comments (optional):

A few typing errors need to be corrected.

Reviewers' comments:

Reviewer's Responses to Questions

**Comments to the Author**

1. Does the manuscript provide a valid rationale for the proposed study, with clearly identified and justified research questions?

Reviewer #1: Yes

2. Is the protocol technically sound and planned in a manner that will lead to a meaningful outcome and allow testing the stated hypotheses?

Reviewer #1: Yes

3. Is the methodology feasible and described in sufficient detail to allow the work to be replicable?

Reviewer #1: Yes

4. Have the authors described where all data underlying the findings will be made available when the study is complete?

Reviewer #1: Yes

5. Is the manuscript presented in an intelligible fashion and written in standard English?

Reviewer #1: Yes

6. Review Comments to the Author

You may also provide optional suggestions and comments to authors that they might find helpful in planning their study.

Reviewer #1: The authors had addressed the comments posed to them with appropriate modifications to this manuscript.

note on page 52 (track changed copy) it states patients will take tables. Should this be tablets?

page 53: handing - should this be handling?

page 60: "data management staff will be used" should be "will use"

7. PLOS authors have the option to publish the peer review history of their article (what does this mean?). If published, this will include your full peer review and any attached files.

Reviewer #1: No

---

## [Editor Report · Acceptance letter]

2 Jan 2024

PONE-D-23-03350R1 

PLOS ONE

Dear Dr. Hashimoto, 

I'm pleased to inform you that your manuscript has been deemed suitable for publication in PLOS ONE. Congratulations! Your manuscript is now being handed over to our production team.

Kind regards, 

on behalf of

Dr. Gang Qin 

Academic Editor

PLOS ONE